# Antimicrobial Properties of TiNbSn Alloys Anodized in a Sulfuric Acid Electrolyte

**DOI:** 10.3390/ma16041487

**Published:** 2023-02-10

**Authors:** Yu Mori, Satoko Fujimori, Hiroaki Kurishima, Hiroyuki Inoue, Keiko Ishii, Maya Kubota, Kazuyoshi Kawakami, Naoko Mori, Toshimi Aizawa, Naoya Masahashi

**Affiliations:** 1Department of Orthopedic Surgery, Tohoku University Graduate School of Medicine, 1-1 Seiryo-machi, Aoba-ku, Sendai 980-8574, Japan; 2Department of Materials Science, Graduate School of Engineering, Osaka Metropolitan University, 1-1 Gakuen-machi, Naka-ku, Sakai 599-8531, Japan; 3Department of Medical Microbiology, Mycology, and Immunology, Tohoku University Graduate School of Medicine, 1-1 Seiryo-machi, Aoba-ku, Sendai 980-8575, Japan; 4Graduate School of Engineering, Institute for Materials Research, Tohoku University, 2-1-1 Katahira, Aoba-ku, Sendai 980-8577, Japan; 5Department of Radiology, Akita University Graduate School of Medicine, 1-1-1 Hondo, Akita 010-8543, Japan; 6Institute for Materials Research, Tohoku University, 2-1-1 Katahira, Aoba-ku, Sendai 980-8577, Japan

**Keywords:** antimicrobial activity, anodic oxide, photocatalyst, sulfuric acid, TiNbSn alloy

## Abstract

TiNbSn alloy is a high-performance titanium alloy which is biosafe, strong, and has a low Young’s modulus. TiNbSn alloy has been clinically applied as a material for orthopedic prosthesis. Anodized TiNbSn alloys with acetic and sulfuric acid electrolytes have excellent biocompatibility for osseointegration. Herein, TiNbSn alloy was anodized in a sulfuric acid electrolyte to determine the antimicrobial activity. The photocatalytic activities of the anodic oxide alloys were investigated based on their electronic band structure and crystallinity. In addition, the cytotoxicity of the anodized TiNbSn alloy was evaluated using cell lines of the osteoblast and fibroblast lineages. The antimicrobial activity of the anodic oxide alloy was assessed according to the ISO 27447 using methicillin-susceptible Staphylococcus aureus, methicillin-resistant Staphylococcus aureus, and Escherichia coli. The anodic oxide comprised rutile and anatase titanium dioxide (TiO_2_) and exhibited a porous microstructure. A well-crystallized rutile TiO_2_ phase was observed in the anodized TiNbSn alloy. The methylene blue degradation tests under ultraviolet illumination exhibited photocatalytic activity. In antimicrobial tests, the anodized TiNbSn alloy exhibited robust antimicrobial activities under ultraviolet illumination for all bacterial species, regardless of drug resistance. Therefore, the anodized TiNbSn alloy can be used as a functional biomaterial with low Young’s modulus and excellent antimicrobial activity.

## 1. Introduction

Titanium dioxide (TiO_2_) is an n-type semiconductor used as a photocatalyst in air and water purification processes owing to the reactive oxygen species (ROS) generated by the photogenerated charge carriers [1]. When TiO_2_ is irradiated with an ultraviolet (UV) light corresponding to its bandgap energy, the ROS, such as hydroxyl radicals (•OH), superoxide anions, and hydrogen peroxide (H_2_O_2_), are generated due to a reaction between the photogenerated charges and oxygen or water [1]. The ROS were reported to exhibit antimicrobial and antitumor effects by disrupting the structure of the bacteria and tumor cells [2,3,4]. Thus, TiO_2_ coating on the biocompatible Ti alloys may be a promising technology for biomaterials because of its biocompatibility and antimicrobial activity.

In orthopedic surgery, titanium, titanium alloys, stainless steels, and Co-Cr-Mo alloys have been used as implant materials. Although these metals are excellent in terms of biosafety, strength, and corrosion resistance, there are problems to be solved, such as discrepancies in Young’s moduli with bone. The commonly used Ti6Al4V alloy in orthopedic implants exhibits a Young’s modulus of 110 GPa, higher than that of the human cortical bone (10–30 GPa) [5]. In clinical practice, the difference in the Young’s moduli of implants and the human cortical bone leads to a disproportionate stress distribution, resulting in pain and bone atrophy after surgery [6]. To resolve this problem, new titanium alloys, such as TiMoZrTa, TiMoZrSi, and TiZrTaSn alloys, have been developed that possess a low Young’s modulus in combination with biocompatibility and strength, and they have shown excellent material properties and favorable results in preclinical studies have been reported [7,8,9,10,11]. In our institute, the development of a near-β-type TiNbSn alloy with a low Young’s modulus of <50 GPa was speculated to reduce stress shielding and thigh pain [12]. Annealing at temperatures >673 K increased the Young’s modulus of the TiNbSn alloy because of an inverse transformation to α-phase or ω-precipitation [13]. A clinical trial of a hip prosthesis using the TiNbSn alloy indicated its efficacy in reducing thigh pain and stress-shield-induced bone atrophy even three years after surgery [14]. Furthermore, a hip prosthesis made of TiNbSn alloy has been clinically applied in Japan. In addition, preclinical studies reported that intramedullary nails and plates made with the TiNbSn alloy accelerated the osteosynthesis of tibial fractures in mice and rabbits compared to those made with Ti6Al4V alloy or stainless steel [15,16,17,18]. To improve the biocompatibility of TiNbSn alloys, a technique for TiO_2_ deposition by anodic oxidation was developed. Anodic oxides, prepared using the electrolytes of acetic or sulfuric acid aqueous solutions, improved the osseointegration owing to the formation of hydroxyapatite in an experimental model, thereby demonstrating the efficacy of the surface improvement on the biocompatibility of TiNbSn alloys [19,20,21,22,23]. Furthermore, the photocatalytic and antimicrobial activities under UV illumination were confirmed in the anodized TiNbSn alloys, prepared in an electrolyte of sodium tartrate aqueous solution [24,25]. In previous studies, clinical and preclinical studies have shown that the low Young’s modulus of TiNbSn alloys improves stress shielding, which is useful in reducing bone atrophy and promoting bone healing. The usefulness of TiO_2_ coating by anodic oxidation of TiNbSn alloys was investigated regarding the improvement of osseointegration and antimicrobial performance. A method of anodic oxidation to obtain TiO_2_ coating that achieves both osseointegration and antimicrobial performance has not been established. In addition, the crystallinity and thickness of the resulting TiO_2_ coating may also change if the base material of the titanium alloy is changed. In a previous study, the comparison of the anodic oxidation of TiNbSn and Ti6Al4V alloys in sodium tartrate electrolytes showed that the anodized TiNbSn alloy had better wear resistance [26,27]. Different anodic oxidation methods may need to be considered for each base metal to optimize the results.

A few studies reported the antimicrobial activity of the anodized Ti alloys owing to their photocatalytic activities [25,28]. The improvement in the bone conductivity of the anodized TiNbSn alloy, prepared in the sulfuric acid electrolyte, was confirmed [21,22]; however, the antimicrobial activity of the anodized TiNbSn alloy has not been verified yet. Except for anodic oxidation using sodium tartrate, no antimicrobial activity has been reported with anodized TiO_2_. Herein, the antimicrobial activity of the TiNbSn alloy, anodized in the sulfuric acid electrolyte, was investigated based on the crystallinity and electronic band structure of the anodic oxide. The hypothesis of this study is that the high crystallinity and porous microstructure of TiO_2_ obtained by sulfuric acid anodic oxidation under high-voltage conditions are favorable and it may exhibit an antimicrobial effect due to its photocatalytic performance. The structure of coated TiO_2_ will be analyzed to determine whether anatase or rutile structure is more important for photocatalytic performance. Once TiO_2_ anodized by sulfuric acid can exhibit antimicrobial performance, it may be useful as a high-performance titanium alloy surface treatment with osseointegration and antimicrobial properties. Based on the results of this study, it is expected that TiO_2_ coating by anodic oxidation of titanium alloys can improve the biocompatibility and antimicrobial performance of titanium alloys, thereby increasing the usefulness of titanium alloys in biomedical materials.

## 2. Materials and Methods

### 2.1. Preparation of the Anodized TiNbSn Alloy

The TiNbSn alloy with a composition of Ti-21Nb-2Sn (at.%) was fabricated using the extruded and swaged thermomechanical processes, as described in the literature [29], and was provided by Mizuho Co. (Mizuho Co., Tokyo, Japan). A TiNbSn plate with dimensions of 25 mm × 25 mm × 1 mm was polished using an emery paper (1500 grit) and alumina particles (1.0 and 0.3 mm in diameter). Subsequently, the plate was washed in ethanol using an ultrasonicator and used as the anode electrode. Similarly, TiNbSn alloy disk plates with dimensions of 10 mm × 10 mm × 1 mm were prepared for the cytotoxicity analysis. Anodization was performed in 1 M sulfuric acid electrolyte using a 100 mesh Pt electrode with dimensions of 50 mm × 50 mm as the cathode. Galvanostatic control was applied at a constant current density of 50 mA/cm^2^ for 0.5 h using a DC power supply (Matsusada Precision, PRK 500-3.2, Otsu, Japan), as previously described [21,22]. The anodized electrode was rinsed with distilled water, dried at 293 K, and subsequently annealed for 5 h at 723 K in atmosphere.

### 2.2. Surface Analyses

The anodic oxide was characterized as described previously [20,22,24]. The microstructure was observed using a laser microscope (LM, Keyence VK-X 150, Keyence Corporation, Osaka, Japan) and field-emission electron probe microanalyzer (EPMA, JXA-8530F JEOL, JEOL Ltd., Akishima, Japan) at an operating voltage of 15 kV. The crystallographic structure was investigated using an X-ray diffractometer (XRD, PANalytical X’Pert diffractometer, Malvern Panalytical, Almelo, The Netherlands) with Cu Kα radiation (0.15406 nm) at a scan rate of 1°/min and thin-film geometry arrangement at a glancing angle of 0.5° with a rotating detector. The absorption spectra were measured using a UV-visible (vis) spectrophotometer (Jasco V-550, JASCO Corporation, Tokyo, Japan). The specimens obtained after the cytotoxic tests were analyzed using X-ray photoelectron spectroscopy (XPS, Shimadzu, Kratos AXIS-Ultra DLD, Kyoto, Japan) with monochromatic Al Kα radiation at a base pressure of 3.0 × 10^−7^ Pa. The full width at half maximum (FWHM) intensity of the Ag 3d_5/2_ peak was 0.73 eV, and the base pressure of the spectrometer was 6.5 × 10^−8^ Pa. 

### 2.3. Photocatalytic Activity Analyses

The photocatalytic activity of the prepared anodic oxide was investigated via the decomposition of methylene blue (MB) under UV illumination. UV light in the range 340–400 nm with a central wavelength of 365 nm was supplied using a xenon lamp (MAX-35, Asahi Spectra, Tokyo, Japan). The intensity of the radiated light was 1.0 mW/cm^2^ at the surface of the cell comprising the MB solution (3.19 ppm) and anodized alloy. The photocatalytic activity was evaluated by measuring the absorbance of MB at 664 nm using a UV-vis spectrophotometer (Jasco V-550, JASCO Corporation, Tokyo, Japan) [25,30]. The photogenerated hydroxyl radicals (•OH) react with terephthalic acid (TA) in a disodium terephthalate (NaTA) solution, thereby producing 2-hydroxyterephthalic acid (HTA) via a hydroxylation reaction, as shown in the following Equation (1):C_6_ H_4_ (COOH)_2_+ •OH→C_6_ H_4_ (COOH)_2_ OH(1)

The excited light at a wavelength of 315 nm generates HTA and emits fluorescence at a wavelength of 425 nm [31]. A quartz cell, containing 2 mL of TA (2 mM) and anodized oxide, was illuminated using a xenon lamp at an intensity of 1.0 mW/cm^2^, and the fluorescence intensity was measured using a spectrometer (Jasco FP-6200, JASCO Corporation, Japan).

### 2.4. Cytotoxicity Assessment

Cytotoxicity assays were performed in accordance with previous studies [12,22]. Two types of cells, MC3T3-E1 (murine osteoblasts) and L929 (murine fibroblasts) obtained from the RIKEN cell bank, were used for the cytotoxicity assessment. The cells were cultured under conditions of no sample, untreated TiNbSn disks, and anodized TiNbSn disks using 24-well cell culture plates. Each cell line was seeded at 1.0 × 10^4^ cells per well. The MC3T3-E1 and L929 cells were cultured in 1.0 mL of α-minimum essential medium (MEM) comprising 10% fetal bovine serum and antibiotics. Subsequently, the cells were detached using trypsin after 72 h of culture and counted in each well. Experiments were performed with n = 3 for each group.

### 2.5. Antimicrobial Assays

Antimicrobial activity was analyzed according to the evaluation procedure described in the International Organization for Standardization ISO 27447:2019 (Japanese Industrial Standard, JIS R 1702:2012). Gram-positive cocci (methicillin-susceptible Staphylococcus aureus (MSSA; NBRC12732) and methicillin-resistant Staphylococcus aureus (MRSA; ATCC43300)) and Gram-negative bacillus (Escherichia coli (*E. coli*; NBRC3972)) were used in the antimicrobial assays. Each bacterium was incubated on the nutrient agar medium (Difco nutrient agar, Becton Dickinson, Lake Franklin, NJ, USA) at 35 °C for 36–43.5 h. The incubated bacteria were prepared in a nutrient broth medium (Nutrient broth, Eiken Chemical, Tokyo, Japan) with a concentration of 1:500 to achieve a bacterial count of 5.3 × 10^6^ cells/mL. The prepared solution was used for the antimicrobial analysis. Glass plates were used as the negative controls, as described in a previous study [25]. Three anodized TiNbSn alloy and three glass plates were used to perform the ISO 27447 antimicrobial assay. The test bacterial solution (37.5 μL, 2 × 10^5^ cells) was inoculated on the anodized TiNbSn alloy and glass plates; the bacterial solution and specimen adhered to the sterile polyethylene film (VF-10, Kokuyo, Osaka, Japan). Glass plates of 1.1 mm thickness (TEMPAX, Schott, Mainz, Germany) were placed on the top of the Petri dish to prevent desiccation. The bacteria were incubated at 25 °C for 8 h under UV illumination (FL 20S Bl-B 20 W, Nippon Electric Company, Tokyo, Japan) at a wavelength of 352 nm. The intensity of the UV light transmitted through the film and glass plate was adjusted to 0.21 mW/cm^2^. In addition, bacteria on each sample were incubated at 25 °C for 8 h in the dark as a control group. After UV illumination, the specimens and films were placed in plastic bottles and washed with 20 mL of soybean casein digestion broth (SCDLP broth, Eiken Chemical, Japan), comprising lectin and polysorbate, to remove the test bacterial solution. 100 μL of the bacteria-washing SCDLP medium solution was diluted with 900 μL of saline solution to prepare a 1:10 dilution of the bacteria-washing solution. 100 µL of bacteria-washing solution and 1:10 diluted washing solution were added to the nutrient agar medium, and subsequently incubated. After 40–48 h of incubation at 35 °C on the nutrient agar medium, the number of colonies was measured and the viable count was determined. The antimicrobial activity (R_L_) of the TiNbSn alloy and the effect of the UV light illumination (∆R) were determined using the following equations [25]:R_L_ = log_10_ (G_L_⁄T_L_)(2)
∆R = log_10_ (G_L_⁄T_L_) − log_10_ (G_D_⁄T_D_)(3)
where T_L_ and T_D_ are the average viable bacterial counts on the three TiNbSn alloy plates after UV illumination and after storage in the dark, respectively, for 8 h. G_L_ and G_D_ are the average viable bacterial counts on the three glass plates after UV illumination and after storage in the dark, respectively, for 8 h.

The antimicrobial activity was defined as 2.0 or higher following the ISO 27447 (JIS R 1702). When no viable bacteria were observed on the anodized TiNbSn alloy group, the viable bacterial count (T_L_) was set to 10.

### 2.6. Statistical Analyses

Statistical analyses were performed using JMP, version 16 (SAS Institute, Cary, NC, USA). Statistical significance and post hoc analyses of the cytotoxicity tests were performed using one-way analysis of variance (ANOVA) and the Tukey–Kramer test, respectively. Values < 0.05 were considered statistically significant.

## 3. Results

### 3.1. Surface Analyses

Figure 1 shows the variations in the electrode voltage, current density, and electrolyte temperature with respect to the elapsed time during anodization. The galvanostatic control is well achieved, and the voltage increases monotonously with the anodization time. During the anodization, a spark discharge is observed on the surface of the electrode a few minutes after the start. The electrolyte temperature increases with time, despite setting a chiller at 10 °C.

Figure 2 demonstrates the (a) LM and (b,c) SEM images of the anodic oxides. Colonies of several tens of microns in size and rough surfaces are observed in Figure 2a. The surface roughness and ratio of the surface areas are 1.415 μm and 3.36, respectively. The scanning electron microscopy (SEM) images (Figure 2b,c) reveal granular-like structures with submicron pores.

Figure 3 shows a backscattered electron image of anodized TiNbSn and the chemical composition analyzed using WDX (wavelength dispersive X-ray spectrometry) of the circled points in the image. Alloying elements of Nb and Sn were detected along with Ti. Abundant oxygen is attributed to metal oxides and adsorbed OH- from H_2_O molecules, which is later presented through XPS analysis.

Figure 4 shows a thin-film X-ray profile of the anodic oxide. The notation at the top of the XRD profile represents the Miller index and the anodic oxide phase. The symbols A and R stand for anatase and rutile, respectively. Rutile-structured TiO_2_ appears predominantly, with a small fraction of anatase TiO_2_. The primary diffraction of the rutile oxide corresponds to the (101) plane, which is different from that of the reference (ICDD 01-073-2224). In addition, the results are different from the anodized oxide prepared in the sodium tartrate electrolyte, where a major peak was observed corresponding to the 101 plane [27].

Figure 5a demonstrates the absorption spectrum of the anodic oxide. An abrupt decrease in the absorbance is observed at approximately 400 nm, suggesting the occurrence of a band transition of the excited charge carriers. To evaluate the bandgap energy, the absorbance spectrum was transformed through the Kubelka–Munk operation using Equation (4):(4)$$\alpha \times h\nu =k\times {\left(h\nu -E_{g}\right)}^{n}$$
where 𝛼, *h*, *v*, and *k* are the absorption coefficient, Planck’s constant, wave number, and a constant, respectively. n is a constant that depends on the electron transition under irradiation. The rutile TiO_2_ exhibits a direct transition of the photogenerated electrons from the conduction band to the valence band [32]. The constant *n* = 0.5 was adopted from the literature [33]. Figure 5b is a plot of the Kubelka–Munk transformation of the absorption spectrum of the anodic oxide. The bandgap energy is calculated as 2.96 eV, which is consistent with the 3.0 eV value given in the literature [34].

Figure 6 shows the XPS of the anodic oxides (a) Ti 2p, (b) Nb 3d, (c) Sn 3d, (d) O 1s, (e) S 2p, and (f) Ca 2p after the cytotoxic analysis. Two individual peaks at approximately 464.2 eV and 458.4 eV in Figure 6a correspond to the Ti 2p_1/2_ and Ti 2p_3/2_ contributions of TiO_2_. The XPS of the Nb 3d (Figure 6b) exhibits peaks at approximately 209.5 eV and 206.7 eV, which correspond to the Nb 3d_3/2_ and Nb 3d_5/2_ contributions, respectively, of Nb_2_O_5_. The peaks (Figure 6c) at 494.6 eV and 486.1 eV suggest the Sn 3d_3/2_ and Sn 3d_5/2_ contributions, respectively, of SnO or SnO_2_. The asymmetrical shape of the O 1s XPS (Figure 6d) is ascribed to metal oxide at approximately 529.7 eV and hydroxyl radicals at 531.3 eV. Sulfur originates from the sulfuric acid in the electrolyte; therefore, the XPS of S 2p (Figure 6e) exhibits peaks of sulfate S 2p_3/2_, sulfate S 2p_1/2_, and sulfide S 2p at 168.1, 169.3, and 163.6 eV, respectively. Spectra deconvolution (Figure 6d,e) was performed using Casa XPS software (www.casaxps.com, (accessed on 5 November 2022)). The XPS spectrum of the Ca 2p (Figure 6f) exhibits two individual peaks at 350.6 eV and 347.1 eV, representing Ca 2p_1/2_ and Ca 2p_3/2_, respectively.

### 3.2. Photocatalytic Assessment

The MB decomposition reaction was fit linearly to a pseudo-first-order kinetic model based on the Langmuir–Hinshelwood kinetic model [35]:(5)$$r=\frac{dC}{dt}=\frac{kKC}{1+KC}\approx k_{\mathrm{app}}C$$
where *r* is the rate of MB degradation, *C* is the absorbance of MB at 664 nm, *t* is the illumination time, *k* is the reaction rate constant, and *K* is the reactant adsorption coefficient. The apparent reaction rate constant *k*_app_ is calculated as 0.00352 min^−1^ (Figure 7); *C_o_* is the concentration of MB before illumination.

Figure 8 shows the fluorescence spectra of the anodic oxide. The intensity at 425 nm is observed >800, which is higher than that for the anodic oxide on pure Ti prepared in the aqueous electrolyte containing sodium tartrate dihydrate and hydrogen peroxide [28].

### 3.3. Cytotoxic Assessment

The cytotoxic analyses of the osteoblast-based cell line culture without samples and in the presence of the untreated TiNbSn alloy disks and sulfate-anodized TiNbSn alloy disks indicate that osteoblast cells proliferate without a significant difference in each group of both the MC3T3-E1 and L929 cells, suggesting that the sulfate-anodized TiNbSn alloy is not cytotoxic, similar to the untreated TiNbSn alloy (Figure 9 and Table 1).

The values are expressed as mean values and standard deviations.

### 3.4. Antimicrobial Assays

Antimicrobial assays were performed in duplicates for each bacterial strain according to ISO 27447. Table 2 lists the antimicrobial activity of the anodized TiNbSn alloy under UV light illumination at an intensity of 0.21 mW/cm.

The antimicrobial activities of the anodic oxide under UV illumination are >2.0 for all bacterial strains. Regardless of the types and structures of bacteria, the anodized TiNbSn alloy exhibits antimicrobial activity under low-intensity UV illumination. The anodized TiNbSn alloys demonstrate antibacterial activities against the drug-susceptible MSSA and drug-resistant MRSA as well (Figure 10).

## 4. Discussion

Herein, we demonstrated the photocatalytic and antimicrobial activities of the anodized TiNbSn alloy, prepared in the sulfuric acid electrolyte. The cytotoxicity was not observed in the TiNbSn alloy after anodization. The surface of the anodized TiNbSn alloy exhibited a uniform porous structure with submicron-sized pores; the rutile-structured TiO_2_ phase was predominant.

In recent studies, when high voltage was applied to an electrolyte of sodium tartrate with or without H_2_O_2_, the anodic oxide formed a porous microstructure independent of the addition of H_2_O_2_ [24,25]. These results are consistent with those of the present study. The high voltage during anodization was speculated to induce dielectric breakdown, thereby resulting in a spark discharge. The spark discharge generated Joule heating on the electrode, leading to the formation of high-temperature rutile TiO_2_ with high crystallinity and porous microstructure owing to the oxygen ion incorporation and micro-dissolution. Since H_2_O_2_ is a strong oxidant, the addition of H_2_O_2_ to sodium tartrate, which is a weak acid, is speculated to accelerate the anodic oxidation reaction and crystallization with low lattice defects, which function as recombination sites for the photogenerated carriers. This was confirmed by the photocatalytic activity and the amount of the generated •OH [24]. Conversely, the anodic oxidation in the sulfuric acid electrolyte promoted the oxidation reaction even without any oxidizing additives because of the strong acidity of the sulfuric acid, resulting in a highly crystallized porous oxide. We believe that the acceleration of the anodic oxidation reaction in a strong acid electrolyte leads to an increase in the crystallinity of TiO_2_, which is responsible for the photoinduced characteristics.

The XRD analysis suggested that the anodic oxide, prepared in the sulfuric acid electrolytes, induced the formation of rutile-structured TiO_2_. The lifetime of the photogenerated charge carriers in the anatase was reported to be longer than that in the rutile, owing to the direct and indirect electron transition structure [32,36,37] in the rutile and anatase, respectively [38]. Therefore, TiO_2_ with an anatase structure showed better photocatalytic activity than that with a rutile structure. Furthermore, TiO_2_ with an anatase structure was reported to exhibit superior photocatalytic and antimicrobial activities to those of the rutile structures [32,36,37]. In contrast, our study exhibited better photocatalytic and antimicrobial activities of the anodic oxide with a rutile structure. This was attributed to high crystallinity that reduced the lattice defects and suppressed the recombination of the photogenerated charge carriers, generated because of the Joule heating induced by a dielectric breakdown.

The ISO 27447 antimicrobial assays exhibited antimicrobial activity of the anodized TiNbSn alloy, prepared in the electrolyte of sulfuric acid aqueous solution, against the MSSA, MRSA, and *E. coli* under low-intensity UV illumination, indicating that the photocatalytic activities of the anodic oxides of the TiNbSn alloys were adequate to kill pathogenic bacteria, regardless of drug resistance. Furthermore, the improved antimicrobial activities can offer potential clinical applications. The proposed anodic oxidation method may contribute towards curing intractable bacterial infections such as vancomycin-resistant bacterial infection [39]. Although antimicrobial agents have side effects, the TiO_2_ surface is considerably safe because the anodic oxide adheres strongly to the substrate and TiO_2_ is biologically inert [40,41]. The antimicrobial effect of the anodic oxidation is reasonably safe. The antimicrobial activity of the anodic oxide, prepared in sodium tartrate electrolytes, has already been reported [25], and the results of our study showed comparable antimicrobial activity to that of the previous report. Furthermore, the sulfate-anodized TiNbSn alloys were reported to have excellent bone affinity owing to the in vivo formation of hydroxyapatite [21], which may facilitate excellent osteoconductive and antimicrobial activities to the functional materials. This study is the first step toward imparting antimicrobial activity using the anodic oxidation of Ti alloys for biological use. When applied to orthopedic implants, it is desirable to have antimicrobial performance even after the installation. Therefore, antimicrobial activity can be imparted in addition to the photocatalytic activity of the anodized TiNbSn alloys in the application of TiNbSn alloys to hip and knee joint replacement prostheses by loading an active antimicrobial substance onto a porous anodic oxide film [42,43]. The development of anodic oxidation techniques for the TiNbSn alloys is expected to lead toward developing biomedical materials with antimicrobial activities.

There are other reports on biocompatible titanium alloys with a low Young’s modulus besides TiNbSn alloys [7,8,9,10,11]. TiNbSn alloys have been clinically applied in hip replacement stems. On the other hand, there are currently no low Young’s modulus titanium alloys with clinical applications in orthopedics other than TiNbSn alloys [14]. The authors expect that the clinical application of low Young’s modulus titanium alloys will improve clinical outcomes in arthroplasty by overcoming the problems of implant strength and production cost. As well as low Young’s modulus, osteoinductive and antimicrobial performance are also expected factors for orthopedic implants [44]. In this study, we demonstrated the antimicrobial effect of by anodic oxidation of TiNbSn alloy in a sulfuric acid electrolyte. In a previous study, we also demonstrated the improvement of osseointegration of TiNbSn alloys anodized in a sulfuric acid electrolyte [21,22]. Anodic oxidation of titanium alloys has been reported to improve corrosion resistance and biocompatibility [45,46,47,48]. The anodic oxidation technique is expected to be a surface processing technology that can improve the function of titanium alloys. In future studies, it is necessary to develop an anodization technique using large specimens to apply the sulfuric acid anodization technique examined in this study on orthopedic implants.

Anodic oxidation of TiNbSn alloy in a sodium tartrate electrolyte has also reported antimicrobial performance using its photocatalytic performance [25]. Anodic oxidation with sodium tartrate has been reported to give TiNbSn alloys excellent wear resistance [26]. In a study comparing the wear resistance of TiNbSn and Ti6Al4V alloys anodized in a sodium tartrate electrolyte, the TiNbSn alloys coated with TiO_2_ by anodic oxidation showed superior wear resistance [27]. TiNbSn alloys may be superior titanium alloys as substrates for anodic oxidation. The anodic oxidation of other titanium alloys, such as Ti6Al4V alloys, has also been reported to improve osseointegration, corrosion resistance, and biocompatibility [45,46,47,48]. Currently, only sulfuric acid anodization under high-voltage conditions has achieved both improved bone affinity and antibacterial performance, and the development and advancement of other anodization techniques is expected in the future. Anodic oxidation technology is promising as a surface processing technology that can improve the function of titanium alloys.

There are several limitations in this study. Anodic oxidation was performed on small disk specimens. In the future, it will be necessary to examine whether it is possible to apply a uniform coating of TiO_2_ with high crystallinity when anodizing a large sample, such as hip replacement prostheses. Since the wear and corrosion resistance of the samples have not been investigated, this will be a subject for future study.

## 5. Conclusions

The photocatalytic activity of the anodized TiNbSn alloy, prepared in the sulfuric acid electrolyte, was demonstrated. A porous microstructure and well-crystallized rutile TiO_2_ with a small amount of anatase structure were observed. The anodic oxide exhibited a robust antimicrobial activity under UV illumination. The results of this study suggest that TiO_2_ coating of titanium alloys by anodic oxidation is expected to improve the biocompatibility and antimicrobial performance of titanium alloys, thereby increasing their usefulness as biomedical materials. The anodized TiNbSn alloy in the sulfuric acid electrolyte may be a promising biomaterial because of its low Young’s modulus and excellent antimicrobial properties, resulting from the oxide crystal and electronic band structure.

## Figures and Tables

**Figure 1 materials-16-01487-f001:**
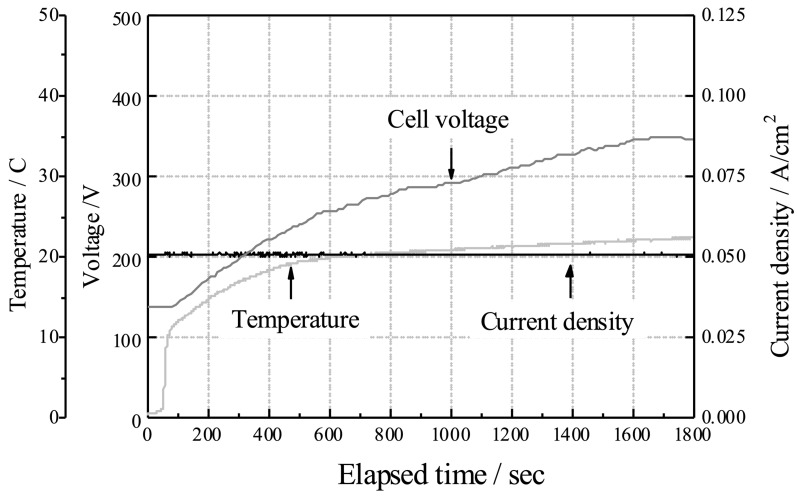
Variation in the voltage and temperature over time during anodization on TiNbSn substrate.

**Figure 2 materials-16-01487-f002:**
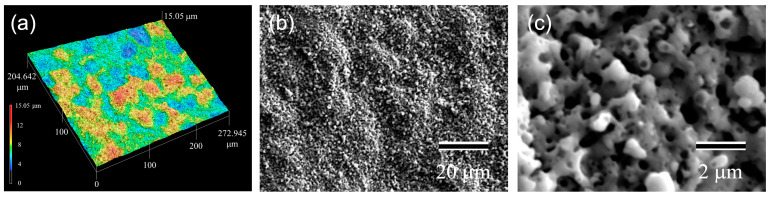
(**a**) LM and (**b**,**c**) SEM images of the anodic oxide.

**Figure 3 materials-16-01487-f003:**
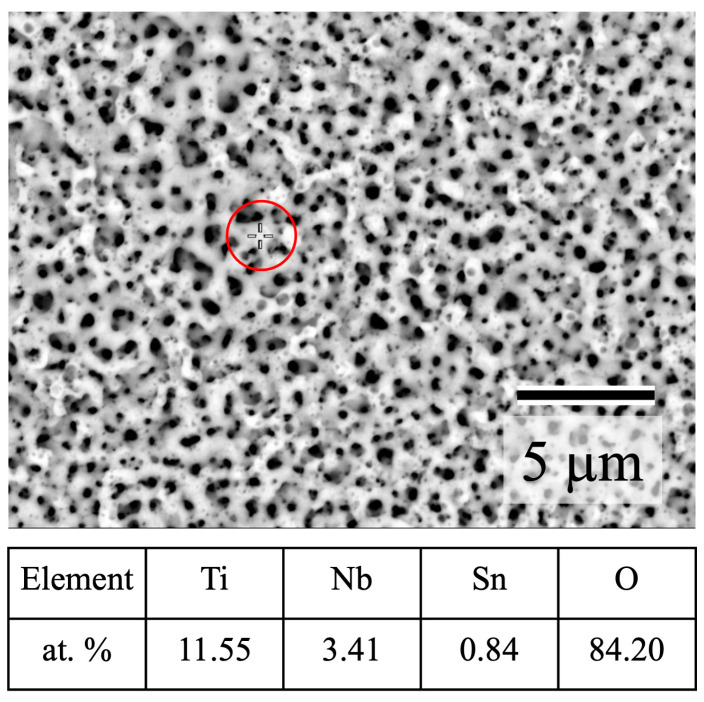
Backscattered electron image of anodized TiNbSn and chemical composition analyzed using WDX of the circled points.

**Figure 4 materials-16-01487-f004:**
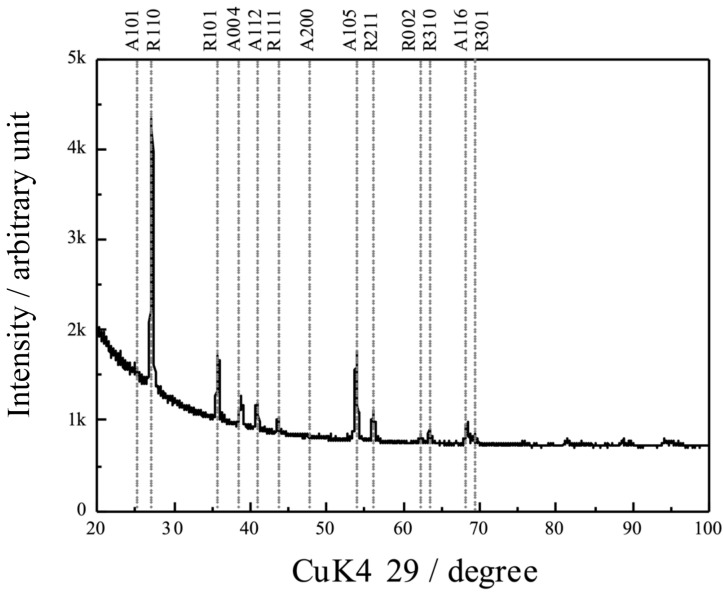
Thin-film XRD scan profiles of the anodic oxides on TiNbSn.

**Figure 5 materials-16-01487-f005:**
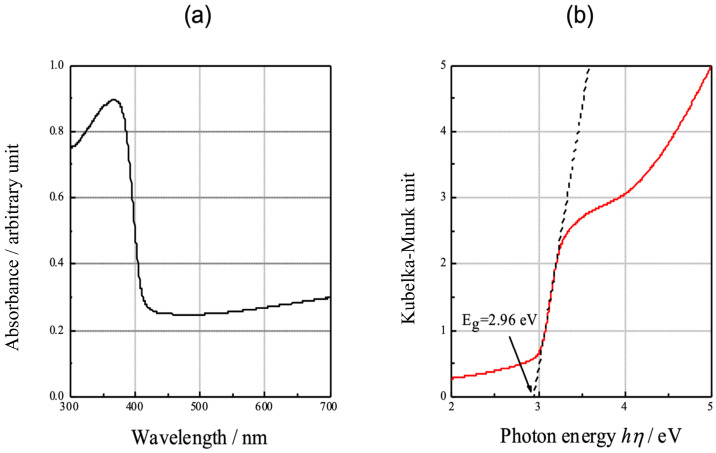
The absorption spectra measured using a UV–vis spectrophotometer. (**a**) Diffused absorption spectra and appearance of anodic oxides. (**b**) Plots of the Kubelka–Munk transformation of the absorption spectrum of the oxide.

**Figure 6 materials-16-01487-f006:**
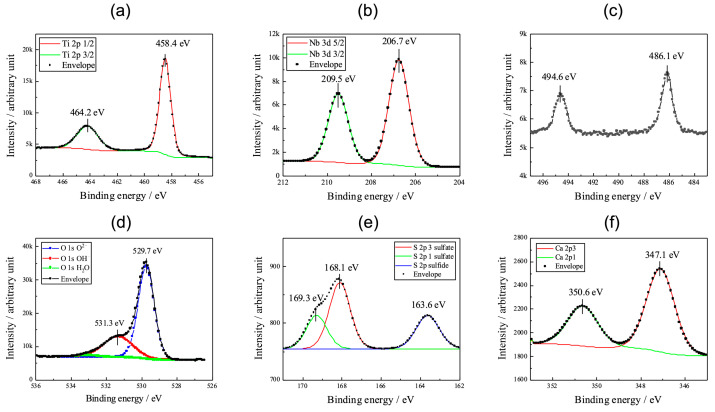
XPS spectra of Ti 2p, Nb 3d, Sn 3d, O 1s, S 2p, and Ca 2p in the anodic oxide. (**a**) Ti 2p, (**b**) Nb 3d, (**c**) Sn 3d, (**d**) O 1s, (**e**) S 2p, and (**f**) Ca 2p.

**Figure 7 materials-16-01487-f007:**
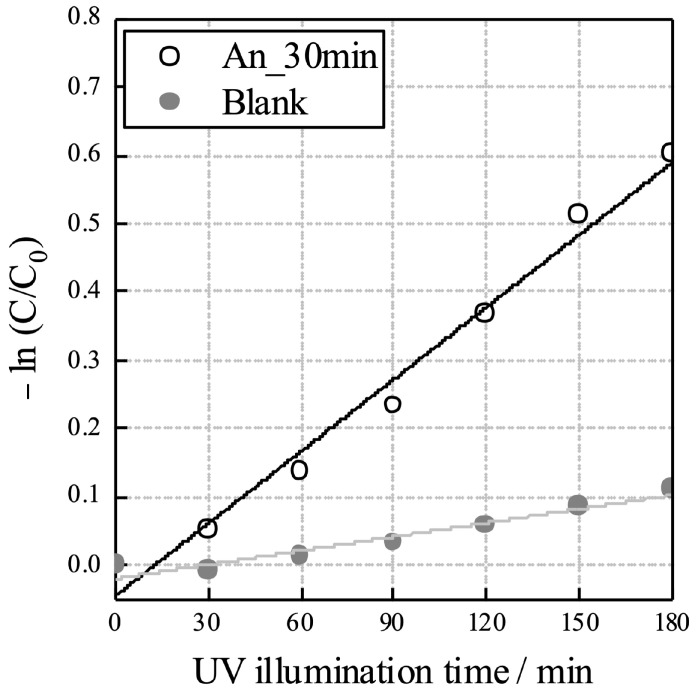
Photocatalytic activity of the anodic oxides on TiNbSn based on the decomposition of MB with the variation in the UV illumination time. The photocatalytic activity was evaluated by measuring the absorbance of MB at 664 nm. The apparent reaction rate constant *k*_app_ is calculated as 0.00352 min^−1^. (AN: annealed, MB: methylene blue, UV: ultraviolet).

**Figure 8 materials-16-01487-f008:**
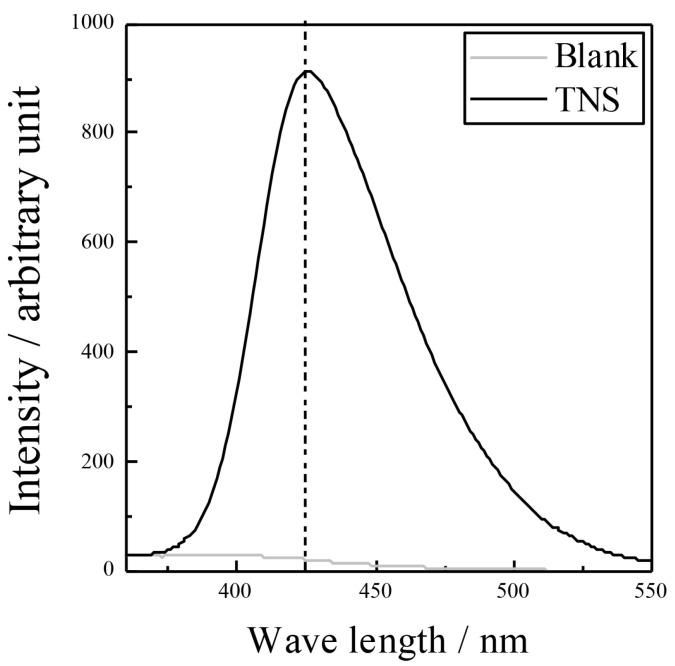
Fluorescence spectra of the anodic oxide. The excited light at a wavelength of 315 nm generated HTA and emitted fluorescence at a wavelength of 425 nm. (HTA: 2-hydroxyterephthalic acid; TNS: TiNbSn alloy).

**Figure 9 materials-16-01487-f009:**
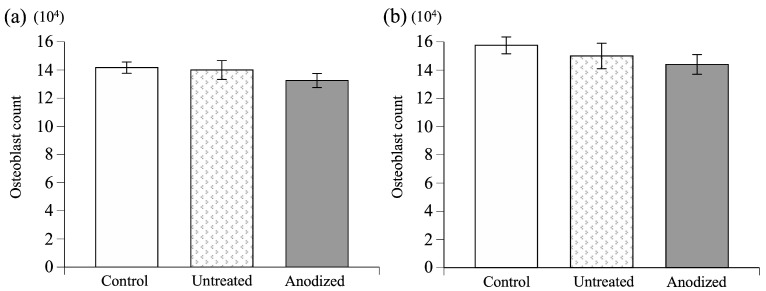
The comparison of cytotoxicity between untreated and anodized TiNbSn alloys. (**a**) Results of cell culture test using MC3T3-E1 cells. (**b**) Results of cell culture test using L929 cells. No significant difference was observed between these groups. One-way ANOVA was used for statistical studies. (ANOVA: analysis of variance).

**Figure 10 materials-16-01487-f010:**
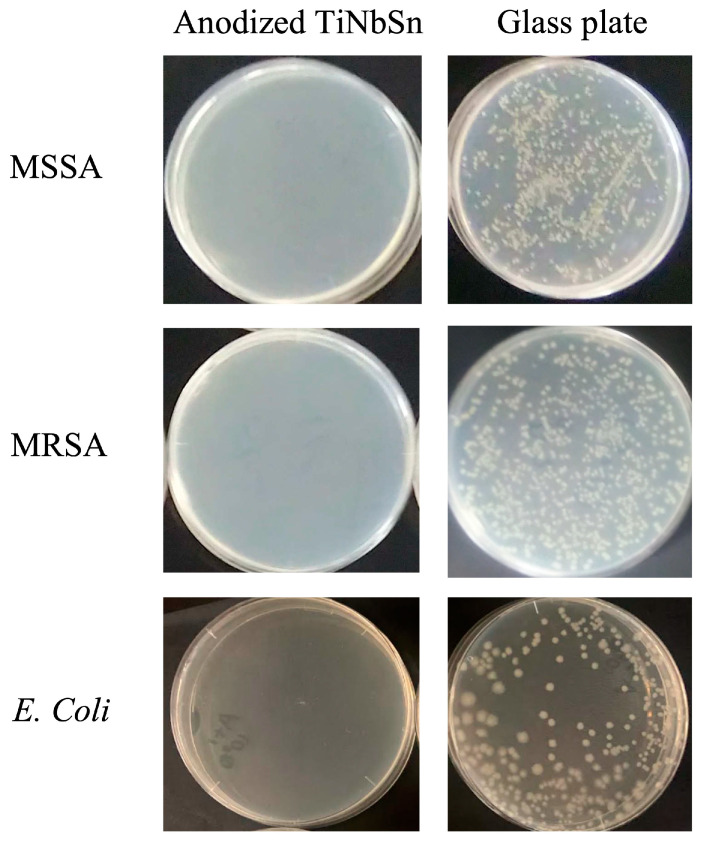
Photographs of the results of antimicrobial assays on anodized TiNbSn alloys and glass plates. (MSSA: methicillin-sensitive Staphylococcus Aureus, MRSA: methicillin-resistant Staphylococcus Aureus, *E. coli*: Escherichia coli.

**Table 1 materials-16-01487-t001:** The comparison of cytotoxicity between untreated and anodized TiNbSn alloys.

Samples	MC3T3-E1	L929
Control	14.2 ± 0.4 (10^4^)	15.8 ± 0.5 (10^4^)
Untreated TiNbSn alloy	14.0 ± 0.7 (10^4^)	15.0 ± 0.5 (10^4^)
Anodized TiNbSn alloy	13.3 ± 0.5 (10^4^)	14.4 ± 0.9 (10^4^)

**Table 2 materials-16-01487-t002:** The assessment of antimicrobial activities and effects of photocatalysis (ISO 27447).

Bacteria	Antimicrobial Activity	Effect of Photocatalysis
MSSA (NBRC12732)	4.16	3.47
MRSA (ATCC43300)	3.58	2.25
*E. coli* (NBRC3972)	4.00	3.76

The assessment was performed in duplicate per assay and repeated twice. Representative data are shown. (MSSA: methicillin-susceptible Staphylococcus Aureus, MRSA: methicillin-resistant Staphylococcus Aureus, *E. coli*: Escherichia coli).

## Data Availability

All data generated or analyzed during this study are included in this published article.

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
