# Peer review of "Antimicrobial Properties of TiNbSn Alloys Anodized in a Sulfuric Acid Electrolyte"

_materials, 2023, doi:10.3390/ma16041487_

Round 1

Reviewer 1 Report

·       Discuss more about practical applications compared with the literature.

·       Add more information in introduction about all alloys used in medical applications.

·       Why you choose this system of alloys?

·       In introduction, you can mention a short comparation of modern titanium alloys versus Ti6Al4V (advantages, disadvantages etc.). The following references are suggested: [1] https://doi.org/10.3390/bioengineering9110686; [2] https://doi.org/10.3390/ma14226806

·       Add more recent references (2021-2022)

·       In the Introduction section, the authors cited the specific results of previous research and cited them adequately. However, they did not mention their shortcomings in previous research. In the Introduction section, the penultimate paragraph should contain common features of previous research. The shortcomings of previous research should also be pointed out, in general.

·       In the Introduction section, the last paragraph should contain the scientific contribution and scientific hypotheses of your research. Complete, further elaborate the scientific contribution and scientific hypotheses of your research. Be explicit. In addition to the goal of the research (which was written), the novelty in the context of the scientific contribution should be pointed out. Scientific contributions should be written based on the shortcomings of previous research in the literature. In this way, the authors will better emphasize novelty and scientific soundness.

·       Specify the supplier, where the materials were purchased.

·       Add a table of EDS test for alloys after obtain.

·       On XRD diffractograms put the Miller indices and the phases. Discuss about the predominant phase on the alloy (Beta or alpha). Compare with the literature.

·       On Cytotoxic assessment, add a table with the values obtained. You can compare with other alloys [3] https://doi.org/10.3390/ma14247610 , [4] https://doi.org/10.3390/mi13030430

·       The study also needs to have a discussion where the authors need to discuss their findings and compare them with the previous studies, either review or empirical ones.

·       The discussion should at least conclude something understandable.

·       In the conclusions, state the scientific contribution, the shortcomings of your methodology and future research.

Author Response

Reviewer 1

  1. Discuss more about practical applications compared with the literature.

Response: Thank you for your comment. We described the practical applications of TiNbSn alloy and anodized TiNbSn alloys in comparison with the literatures.

  1. Add more information in introduction about all alloys used in medical applications.

Response: As reviewer’s suggestion, we described the alloys used in medical applications in introduction.

  1. Why you choose this system of alloys?

Response: We originally developed TiNbSn alloy and performed clinical trial of hip replacement prosthesis.

  1. In introduction, you can mention a short comparation of modern titanium alloys versus Ti6Al4V (advantages, disadvantages etc.). The following references are suggested: [1] https://doi.org/10.3390/bioengineering9110686; [2] https://doi.org/10.3390/ma14226806

Response: Thank you for your comment. We described the comparison between modern titanium alloys and Ti6Al4V alloy using suggested references.

  1. Add more recent references (2021-2022)

Response: We added recent references according to reviewer’s comment.

  1. In the Introduction section, the authors cited the specific results of previous research and cited them adequately. However, they did not mention their shortcomings in previous research. In the Introduction section, the penultimate paragraph should contain common features of previous research. The shortcomings of previous research should also be pointed out, in general.

Response: According to the reviewer’s comment, the authors described the commonalities of previous studies and also added descriptions of issues that have not been accomplished.

  1. In the Introduction section, the last paragraph should contain the scientific contribution and scientific hypotheses of your research. Complete, further elaborate the scientific contribution and scientific hypotheses of your research. Be explicit. In addition to the goal of the research (which was written), the novelty in the context of the scientific contribution should be pointed out. Scientific contributions should be written based on the shortcomings of previous research in the literature. In this way, the authors will better emphasize novelty and scientific soundness.

Response: Thank you for your comment. We have described our hypothesis for this study in the final paragraph of the introduction. We also added a description of the scientific contributions made by the research results.

  1. Specify the supplier, where the materials were purchased.

Response: We described that the TiNbSn alloy material is obtained from Mizuho Co.

  1. Add a table of EDS test for alloys after obtain.

Response: As reviewer’s suggestion, chemical composition of anodized TiNbSn, analyzed by WDX (wavelength dispersive X-ray spectrometry), was added to the revised manuscript. We used WDX, which has higher energy resolution than EDS. The following statements are added.

“Fig. shows backscattered electron image of anodized TiNbSn and chemical composition analyzed by WDX (wavelength dispersive X-ray spectrometry) of the circled points in the image. Alloying elements of Nb and Sn were detected along with Ti. Abundant oxygen is attributed to metal oxides and adsorbed OH- from H2O molecules, which is later presented by XPS analysis.”

  1. On XRD diffractograms put the Miller indices and the phases. Discuss about the predominant phase on the alloy (Beta or alpha). Compare with the literature.

Response: Thank you for your comment. The notation described on the top of XRD profile (Fig. 4) correspond to the Miller indices and the phases. Following sentence to explain the above was added in the revised manuscript.“The notation at the top of the XRD profile represents the Miller index and the anodic oxide phase. The symbols A and R stand for anatase and rutile respectively.”

  1. On Cytotoxic assessment, add a table with the values obtained. You can compare with other alloys [3] https://doi.org/10.3390/ma14247610 , [4] https://doi.org/10.3390/mi13030430

Response: As reviewer’s suggestion, a table summarizing measured values for cytotoxicity test results was added. Added description of biocompatibility with suggested references.

  1. The study also needs to have a discussion where the authors need to discuss their findings and compare them with the previous studies, either review or empirical ones.

Response: Thank you for your comment. The statements regarding discussion were added citing previous studies on low Young’s modulus titanium alloys and anodic oxidation of titanium alloys.

  1. The discussion should at least conclude something understandable.

Response: According to the reviewer’s suggestion, the conclusion has been modified to include additional information on the originality, challenges, limitations, and scientific contributions of the study for a better understanding.

  1. In the conclusions, state the scientific contribution, the shortcomings of your methodology and future research.

Response: Thank you for your comment. The conclusion has been modified to include additional information on the originality, challenges, limitations, and scientific contributions of the study for a better understanding.

Reviewer 2 Report

-Figure 4 contains captions a and b figures, but in Figure 4 there is only 1 figure. Its hard to understand.

-There are studies published by the authors examining the antibacterial properties of such anodized alloys. The originality of this work should be clearly demonstrated in the introduction section.

-Can you give a cross-sectional view of the anodization layer? It is important to characterize and discuss this layer.

-Instead of the expressions A101, R110....in Figure 3, it should be clearly stated which compounds they are.

Author Response

Reviewer 2

  1. Figure 4 contains captions a and b figures, but in Figure 4 there is only 1 figure. It’s hard to understand.

Response: Thank you for your comment. We have added a figure.

  1. There are studies published by the authors examining the antibacterial properties of such anodized alloys. The originality of this work should be clearly demonstrated in the introduction section.

Response: In accordance with the reviewer's remarks, we have described the originality of this study in the introduction.

  1. Can you give a cross-sectional view of the anodization layer? It is important to characterize and discuss this layer.

Response: Thank you for your comment. Unfortunately, we have not conducted cross-sectional observation of the oxide. However, oxides prepared by potentiostatic anodization at 220V have been observed by cross-sectional TEM (Materials Science & Engineering C 98 (2019) 753–763), and we consider the present oxide (the recorded maximum voltage is about 350V) has similar cross-sectional microstructure to the above except for oxide thickness and density of the incorporated pores. An increase in applied voltage promotes anodizing reaction by spark discharge due to dielectric breakdown, and the oxide thickness increases and the number of the incorporated pore increase. A spark discharge occurred at 220V, and was also observed in the current anodization as described in Discussion. Since pores with a size of several hundred nanometers exist as closed pores in the oxide layer, it is thought that the antimicrobial activities are not affected by the pores.

  1. Instead of the expressions A101, R110....in Figure 3, it should be clearly stated which compounds they are.

Response: According to the reviewer’s comment, we have added following sentence in the revised manuscript.

“The notation at the top of the XRD profile represents the Miller index and the anodic oxide phase. The symbols A and R stand for anatase and rutile respectively.”

Round 2

Reviewer 1 Report

Paper was improved.

Reviewer 2 Report

Congratulations to the authors for their work.